# Assessing Metabolic Markers in Glioblastoma Using Machine Learning: A Systematic Review

**DOI:** 10.3390/metabo13020161

**Published:** 2023-01-21

**Authors:** Zachery D. Neil, Noah Pierzchajlo, Candler Boyett, Olivia Little, Cathleen C. Kuo, Nolan J. Brown, Julian Gendreau

**Affiliations:** 1School of Medicine, Mercer University, Savannah, GA 31404, USA; 2Department of Neurosurgery, Jacobs School of Medicine and Biomedical Sciences at University at Buffalo, Buffalo, NY 14203, USA; 3Department of Neurosurgery, University of California Irvine, Orange, CA 92697, USA; 4Department of Biomedical Engineering, Johns Hopkins Whiting School of Engineering, 3400 N Charles St., Baltimore, MD 21218, USA

**Keywords:** glioblastoma, metabolism, biomarker, machine learning, artificial intelligence, deep learning, diagnosis, prognosis

## Abstract

Glioblastoma (GBM) is a common and deadly brain tumor with late diagnoses and poor prognoses. Machine learning (ML) is an emerging tool that can create highly accurate diagnostic and prognostic prediction models. This paper aimed to systematically search the literature on ML for GBM metabolism and assess recent advancements. A literature search was performed using predetermined search terms. Articles describing the use of an ML algorithm for GBM metabolism were included. Ten studies met the inclusion criteria for analysis: diagnostic (n = 3, 30%), prognostic (n = 6, 60%), or both (n = 1, 10%). Most studies analyzed data from multiple databases, while 50% (n = 5) included additional original samples. At least 2536 data samples were run through an ML algorithm. Twenty-seven ML algorithms were recorded with a mean of 2.8 algorithms per study. Algorithms were supervised (n = 24, 89%), unsupervised (n = 3, 11%), continuous (n = 19, 70%), or categorical (n = 8, 30%). The mean reported accuracy and AUC of ROC were 95.63% and 0.779, respectively. One hundred six metabolic markers were identified, but only EMP3 was reported in multiple studies. Many studies have identified potential biomarkers for GBM diagnosis and prognostication. These algorithms show promise; however, a consensus on even a handful of biomarkers has not yet been made.

## 1. Introduction

Glioblastoma multiforme (GBM) is the most common primary malignant brain tumor in the United States, accounting for approximately 56.6% of all gliomas and 47.7% of all primary malignant central nervous system tumors [1]. GBM is 1.58 times more common in males than females, and the annual incidence of GBM is 2.53 per 100,000 population. The highest rate of diagnosis falls with the group aged 75 to 84 years; however, the median age of diagnosis is 65 years. Globally, the incidence is highest in North America, Northern and Western Europe, and Australia [2]. When accounting for race and ethnicity, incidence rates are highest among non-Hispanic whites and lowest among American Indians and Alaskan Natives. Furthermore, 1- and 5-year survival rates are lowest among non-Hispanic whites and highest among American Indians and Alaskan Natives [3].

The prognosis of GBM is notably grim, with a 1-year relative survival rate of 41.4% and a 5-year survival rate of 5.8% following diagnosis [1,3,4]. Negative prognostic factors include advanced age, incomplete resection, and poor mental performance status while the inverse of these factors each indicates a slightly better prognosis [4]. Furthermore, biomarkers indicating isocitrate dehydrogenase 1 (IDH1) and IDH2 mutations indicate a longer survivability [5].

Currently, no screening method for GBM prior to clinical presentation exists. Only once clinical symptoms are present does magnetic resonance imaging (MRI) become the gold standard for diagnosis [6]. This lack of reliable screening leads to diagnoses late into the progression of cancer. The development of techniques leading to early detection of GBM may play an integral role in improving patient outcomes following diagnosis. Current research suggests that early interventions with surgical resection, radiation therapy, and pharmacological targeting of the neoplasm may improve patient outcomes [7]. Additional research suggests that early resection of the tumor may play a role in preventing disease progression as GBM tumors display rapid early progression (REP) indicating that early phases of tumor growth are crucial to the growth of the neoplasm [8]. The identification of metabolic biomarkers such as IDH, platelet-derived growth factor (PDGF), and epidermal growth factor receptor (EGFR) provides an opportunity for early detection of risk factors and prognostic factors relating to GBM [6].

Traditional cancer diagnoses are determined by a physician via clinical, imaging, and population-based data, with confirmation via histology upon biopsy or autopsy [9]. Recently, machine learning (ML, a subset of artificial intelligence [AI]) is improving the diagnostic and prognostic processes for various cancers [10,11,12,13]. Machine learning is a method of teaching computers to learn from data without explicit programming. Instead, algorithms are fed massive amounts of training data to identify patterns and make classifications and predictions about new, untested data. This is accomplished by creating mathematical models that can learn from existing data and then use these models to predict new data.

The first mention of AI was in 1956 during a seminar at Dartmouth College [14,15]. Decades later, ML was born in the mid-1980s after Valiant’s theory of the learnable (1984) and Hopfield’s neural network model of associative memory (1982) first connected statistical mechanics to learning theory, thereby replacing the previous AI approach centered on logic and rules [16]. The first applications of ML in neurosurgery began in the 1990s [17]. An early proof-of-concept by Floyd et al. in 1992 demonstrated that artificial neural networks (ANN) could outperform human efforts in detecting circular lesions in stimulated single-photon emission CT imagery [17,18]. In 1995, ML was utilized in a study by Christy et al. for grading and distinguishing between high and low-grade supratentorial astrocytomas [17,19]. The results of this study, though nonsignificant, further demonstrated the diagnostic capabilities of ML with an accuracy of 61% when compared to 57% for neuroradiologists [17,19].

Cancer gene expression has been a prominent focus of ML in addition to MRI/CT imaging analysis, cancer susceptibility testing, radiation resistance, mortality risk percentages, and tumor grade [10,20,21,22,23]. The ML techniques commonly utilized within cancer research include ANN, K-nearest neighbors (KNN), Bayesian network (BN), Naïve Bayes (NB), support vector machine (SVM), and decision trees (DT). Of these approaches, ANNs, KNN, and SVMs exhibit popular use among researchers working to ascertain a cancer diagnosis [9,20]. Traditional ML methods including DTs, KNNs, NB, and SVMs are more simplistic, which results in greater computational speed, efficiency, and cost savings [20]. Past attempts to diagnose GMB include a notable two-stage ML-based study that created a multimodel, multichannel predictive model consisting of a convolution neural network (CNN) (a subset of ANN) connected to an SVM. The two systems analyzed MRI images (T1, diffusion tensor imaging, and resting state MRI) along with tumor histology and patient age. This ML model projected preoperative high-grade glioma survival rates that were 90.66% accurate (N = 68) [24].

The exponential advancement of AI has led to the creation of ML models that utilize electronic health record data to achieve a 60% positive predictive value (PPV) for a 3-month mortality rate in an advanced cancer population (N = 2041) prognostic algorithm [25]. The algorithm’s PPV markedly surpassed the 34.8% PPV attained by oncologists and advanced practice clinicians [25]. For diagnostic applications, a study from Zhou et al. incorporated liquid chromatography and mass spectrometry into an SVM-based ML algorithm to diagnose malignant brain gliomas (MBGs) by means of plasma lipid biomarker analysis. This method was shown to be a reliable noninvasive screening method for the diagnosis of MBGs [26]. The enhancement of diagnostic/prognostic methods integrated with ML algorithms allows physicians to assign patients more accurate prognoses for the expeditious implementation of treatment plans and conceivably better patient care as the technology continues to improve [4,10].

The heterogeneous nature of GBM along with high rates of re-incidence and therapeutic resistance necessitate the timely identification of novel therapeutic targets in the metabolism of GBM to remain ahead of this rapidly evolving disease [27]. Recent efforts to identify such targets have utilized tumor omics data integrated with clinical information by use of ML techniques [27]. However, there is still a paucity of literature concerning GBM metabolism and ML. To our knowledge, there has been no review on ML and GBM metabolism. Therefore, in this review, the authors systematically search the literature on ML and GBM metabolism and assess recent advancements with commentary on future developments in this novel and the emerging field of study.

## 2. Methods

### 2.1. Strategy and Registration

This study was performed in accordance with the Preferred Reporting Items for Systematic reviews and Meta-analyses (PRISMA) guidelines. This systematic review was registered on PROSPERO, with details of our initial protocol, and can be accessed at https://www.crd.york.ac.uk/prospero/display_record.php?ID=CRD42022367758 (accessed on 25 December 2022) [28].

### 2.2. Search and Data Sources

A literature search was performed using PubMed (Medline), Embase, Cochrane, OVID, and Web of Science databases from 1975 to October 2022. The following predetermined search terms were used: “metabolism” or “biomarkers” and “glioblastoma” and “artificial intelligence” or “machine learning” or “deep learning” or “predictive model” as title and abstract keywords (Appendix A).

### 2.3. Selection Criteria

Articles obtained from searching the specified databases were imported into the Covidence platform (Veritas Health Innovation) for screening. The screening was independently performed by two investigators (Z.D.N. and C.B.) by title and abstract, and later by full-text review. Conflicts were resolved by consensus. When consensus could not be obtained, a third reviewer (N.P.) broke the tie. Articles describing the use of an ML algorithm for GBM metabolism were included. Additional inclusion criteria included research on human GBM metabolism and a confirmatory diagnosis of GBM for validity. Review articles, case reports, commentary, conference abstracts, unpublished articles, editorials, and purely technical descriptions were excluded. The language was restricted to English.

### 2.4. Data Extraction

Using a data extraction form, a quality and bias assessment was performed based on the quality assessment of diagnostic accuracy studies 2 (QUADAS-2) [29] for diagnostic studies and the quality assessment of prognostic accuracy studies (QUAPAS) [30] for prognostic studies. QUAPAS is adapted from QUADAS-2 and thus, the two can be compared together. Studies that featured both diagnostic and prognostic models were assessed with both assessments and combined. All studies were also assigned a level of evidence rating based on the American Association of Neurological Surgeons (AANS) and Congress of Neurological Surgeons (CNS) joint guidelines for diagnostic and prognostic studies organized into classes (I, II, and III) [31].

The following variables were extracted from the included studies: publication year, lead author, country of origin, study population, the origin of patient data (original or database), type of ML algorithm used, and source of biologic sample (plasma, tissue, etc). Our primary outcome was the accuracy of the ML algorithm. The secondary outcome, if reported, included the top metabolic markers identified by each study.

## 3. Results

### 3.1. Search Results

Our initial search yielded 317 records with 235 articles remaining after the removal of duplicates (Figure 1). These articles were screened by title and abstract, which returned 46 articles for a full-text review. Thirty-one articles were excluded, leaving 14 final studies included in this review, ten of which were included in the analysis [32,33,34,35,36,37,38,39,40,41]. Four studies were not included in the analysis because they did not perform an ML algorithm on GBM; however, they did discuss the topic [21,42,43,44,45,46].

Extraction results from each paper are tabulated below (Table 1). All of the ten analyzed papers were published on or after 2018, except one (2012). The nationalities of the papers were as follows: one Canadian, four Chinese, and one Indian, Pakistani, Polish, Swiss, and American. Three papers (30%) were diagnostic, six were prognostic (60%), and one featured both (10%).

### 3.2. Quality and Bias and Level of Evidence

A quality and bias assessment was performed as described above with results tabulated in the Appendix A. Weakness in the included studies was most significant for the lack of data in the study design. Fifty-one percent (n = 24) of risk of bias questions were answered as “unsure” due to insufficient information (Figure 2). Patient selection was also a substantial source of bias as most studies used some form of a national database with little information provided on the patient population or control population if used. Conversely, 78% (n = 29) of applicability concern questions were rated as “low” risk of bias (Figure 3). Nine studies (90%) received a level of evidence rating of II, while one study (10%) received a rating of I, based on AANS and CNS joint guidelines for diagnostic and prognostic studies.

### 3.3. Patient Samples and Databases

Samples were collected from patients as either tumor/healthy brain tissue (n = 7, 70%) or serum/plasma (n = 3, 30%). Most studies analyzed data from multiple national databases such as The Cancer Genome Atlas (TCGA) (National Institutes of Health) (n = 6, 60%), Chinese Glioma Genome Atlas (CGGA) (Beijing Neurosurgical Institute) (n = 2, 20%), Gene Expression Omnibus (GEO) (National Institutes of Health) (n = 3, 30%), as well as from original samples (n = 5, 50%). However, only three articles (30%) analyzed data from multiple sources. At least 2536 data samples were run through an ML algorithm; however, due to the usage of the same databases by multiple papers at various points of database completeness, an exact number of unique samples could not be determined.

### 3.4. Machine Learning and Accuracy

Twenty-seven ML algorithms were found in our analysis, 18 of which were unique (67%) (Figure 4). The least absolute selection and shrinkage operator (LASSO) and support vector machine (SVM) were the two most common methods and were featured in five and four studies, respectively. A mean of 2.7 ML methods was utilized in each study; however, only 50% (n = 5) of papers featured more than two methods of ML. Of the 27 ML methods used, 24 were supervised (89%) and three were unsupervised (11%), while 19 were continuous (70%) and eight were categorical (30%). A summary of each ML method is listed in (Table 2).

**Table 2 metabolites-13-00161-t002:** A summary of each ML method is listed in these studies. This includes methods that were mentioned; however, some may not have been utilized, but are still included for educational purposes [32,33,34,35,36,37,38,39,40,41]. * Duplicate methods omitted.

Machine Learning Algorithm	Definition
Linear Regression (ACE)	Linear regression is a type of supervised ML algorithm used for predictive modeling. It is used to match observed data with a linear equation to model the correlation between the independent variables and the dependent variable [45]. ACE is a linear regression algorithm specifically designed for use with gene expression data.
Logistic Regression	Logistic regression is a statistical method that we use to construct a regression model when the response variable is in binary. It is integrated into a supervised machine learning algorithm to hypothesize an outcome along with a binary response (e.g., Yes/No, True/False) using a set of independent variables [46].
Random Forest *	Random forest is a supervised ML method that creates decision trees and combines them to improve the accuracy of the predictions. It uses a technique called bagging, where each tree is trained on a random grouping of the data [45].
Extra Tree Classifier	Extra tree classifier is a supervised ML algorithm that utilizes a decision tree-based ensemble method. It operates by constructing a set of decision trees and then training them with a random subset of the features. The final class prediction is created by combining all of the trees’ individual class predictions. Extra tree uses more randomization when splitting nodes than is seen in a random forest algorithm [47].
Decision Tree	A decision tree is a type of supervised ML algorithm that is used for classification and regression. It makes predictions based on the feature values of input instances by constructing a tree-like model of decisions and their possible consequences [45].
SVM (Support Vector Machines) *	Support vector machines is a supervised ML algorithm that is used for classification and regression. It works by finding the best boundary (or “hyperplane”) that separates the different classes [45].
ANN (Artificial Neural Networks) CNN (Convolutional Neural Network) BPNN (Backpropagation Neural Network) DNN (Dropout Neural Network) PASNet (Pathway-Associated Sparse Deep Neural Network)	ANN is a class of supervised ML algorithms that are modeled after human neuronal structure and can be applied to a variety of tasks, including the classification of images and the processing of natural language. They are made up of interconnected artificial neurons that can be trained to adjust the weights of connections between nodes. They can use a variety of architectures, including feedforward, convolutional, and recurrent neural networks [45]. CNN is a type of neural network that is commonly utilized in the recognition of images and videos. It uses convolutional layers to learn spatial hierarchies of features automatically and adaptively from input data [48]. Backpropagation is an ML algorithm for multilayer feedforward artificial neural networks (FFNN). Backpropagation is used to train these networks to produce a desired output for a given input [49]. Dropout is a regularization technique for neural networks, which aims to reduce overfitting by randomly setting a portion of the neurons to zero during training. This helps to avoid overfitting by preventing the network from becoming too specialized for the training set [50]. PASNet is a deep learning algorithm that combines feature selection and neural networks to predict disease-gene associations. It is used to identify the genes that are important for a specific disease, by incorporating information about biological pathways into the prediction process [39].
XGBoost (eXtreme Gradient Boosting)	XGBoost is a gradient-boosting supervised ML algorithm designed to be efficient and scalable. It is used for supervised ML problems, and it can be used for both classification and regression [51].
K-Means	K-Means is an unsupervised ML clustering algorithm that groups similar n-dimensional observations into k clusters, where k is predefined. The algorithm repetitively assigns points to the closest centroid and updates the centroid based on the mean of assigned points [52].
LASSO (Least Absolute Selection and Shrinkage Operator) * LASSO-Penalized Cox Regression Logistic LASSO Random LASSO	LASSO is a supervised regularization method for linear regression models. LASSO’s priority is to decrease the absolute values of the independent variable coefficients toward zero. It helps to prevent overfitting by reducing the model’s complexity [53]. LASSO-penalized Cox regression is a method that combines LASSO regularization with the Cox proportional hazards model. It aims to identify a subset of features that are most important for survival analysis while minimizing overfitting [54]. Logistic LASSO is a regularization method for logistic regression. It is a combination of LASSO and Logistic regression and it aims to identify the subset of features that are most important for predicting binary outcomes while also minimizing overfitting [55]. Random LASSO is a variant of LASSO that uses randomization to improve the feature selection process. It randomly assigns weights to the features before applying LASSO, which can help to reduce the variance of the feature selection results [56].
PCA (Principal Component Analysis) *	PCA is an unsupervised dimensionality reduction technique. The intention is to convert a group of correlated factors into a group of uncorrelated factors. It does this by switching the data to a new coordinate system. The axis then represents the direction of maximum variance in the data [57].
RSF-SRC (Random Survival Forest–Survival Regression and Classification)	RSF-SRC is a potentially unsupervised ML method for predicting the time-to-event (TTE) outcome in survival analysis (other variations may be supervised). It is an extension of the random forest algorithm, can handle censoring and truncation of time-to-event data, and can be used for both regression and classification [58].
PLS-DA (Partial Least-Squares Discriminant Analysis)	PLS-DA is a supervised ML algorithm that is used for classification. It works by finding a group of latent variables, which are linear combinations of the original variables, and that explain the differences between various different classes [59].
Naïve Bayes	Naïve Bayes is a supervised ML algorithm that is used for classification. It makes predictions based on the probability of certain features appearing in each class. It is called “naïve” because it assumes that all features are independent, which may not always be true [45].

Nine algorithms (33%) reported accuracy values and 18 (67%) reported area under the curve of the receiver operating characteristic (AUC of ROC) values, while only 6 (22%) reported neither. Due to the unverifiable nature of unsupervised ML, only accuracy or AUC of ROC values not reported for supervised ML methods were considered “missing.” Only one paper met this criterion [33]. The mean reported accuracy was 95.63% [85.70%, 100.00%], while the mean AUC of ROC was 0.779 [0.590, 1.000].

### 3.5. Metabolic Markers

One hundred six metabolic markers were identified as the top predictive biomarkers from the analyzed studies. Of these, 23 (22%) were used for diagnosis and 83 (78%) were used for prognostication. Only one metabolic marker, EMP3, was reported in multiple studies; all other biomarkers were reported only once in their respective studies.

## 4. Discussion

Despite the innovations within the field of GBM research, prognoses remain poor. The studies within this review aim to improve diagnostic and prognostic accuracy by utilizing novel ML algorithms. Although this field exhibits an extensive level of research, there is a paucity of literature pertaining to the ML algorithms used to identify markers underlying GBM metabolism [42].

### 4.1. Supervised Machine Learning

Supervised ML is broadly used in a predictive scenario where a “ground truth” value can be determined (e.g., a diagnosis of GBM) and the user wishes to identify similar data sets with an unknown “ground truth.” The supervised ML algorithms used by these studies were SVM, random forest, ANN, deep neural networks (e.g., PASnet), DT, NB, partial least-squares discriminant analysis (PLS-DA), logistic regression models, and LASSO-penalized Cox regression analysis [32,34,35,36,37,38,39,40,41,42,43]. A logistic regression model appears to outperform other ML algorithms in classification systems, in this case, the classification of the IDH mutation. The algorithms it outperformed were other supervised ML models such as SVM and random forest models. Specifically, the logistic regression model obtained greater results, which were determined by its performance in determining the AUC of ROC, Bal accuracy, F1 score, precision, recall, and Matthew’s correlation coefficient (MCC) [43].

Supervised ML algorithms are powerful tools in the identification of GBM biomarkers. One study found that by extracting a small amount of peripheral blood (5 µL), a surface-enhanced Raman scattering (SERS) signal-trained supervised ML algorithm was able to distinguish GBM cancer from noncancer without isolating cells. The PLS-DA algorithm exhibited both high sensitivity and specificity. A confirmation test with an ANN validated the previous outcome, and the ANN was crucial in determining the prognosis of the disease [32]. Congruently, Gollapalli et al. used a PLS-DA algorithm to distinguish between GBM patients and healthy controls using predetermined biomarker subsets to discern a high level of classification. Results from this study were confirmed with DT, SVM, and NB algorithms [41].

### 4.2. Unsupervised Machine Learning

Unsupervised ML techniques are generally used when a user wishes to understand and perhaps categorize their data, without knowing their primary data “ground truth.” The unsupervised ML algorithms used by these studies were K-means and an integrated Kernel PCA. Unsupervised ML methods such as K-means have been used to create continuous clustering models. These models use metabolism-related genes to create stratified clusters with calculated similarity distances between GBM samples [37]. Furthermore, deep neural networks (e.g., PASnet) have been integrated into prediction models, along with kernel principal component analysis (KPCA), as methods to forecast prognostic survival analyses from high-throughput data [39]. The literature on deep learning networks in GBM metabolism is sparse, likely due to the complicated methodology involved in the construction of these algorithms. Sometimes, a combination of both unsupervised and supervised ML is useful. Riviere-Cazaux et al. contrasted the heterogeneity between GBM patients based on IDH status and patient identity. The team utilized PCA for an unbiased evaluation of patient groupings, followed by PLS-DA to identify the predictive variables between the groups. This is a quality example of how several variants of different algorithms can work together harmoniously to achieve results [44].

### 4.3. Metabolic Markers

A major prognostic metabolic marker researched throughout these studies is isocitrate dehydrogenase-1 (IDH1), which when correlated with various mutations of that marker, as well as when utilized for the characterization of patients, was found to be overexpressed in both high- and low-grade GBM patients [33,37,40]. The type of IDH1 (wild-type vs. mutation) was found to impact the degree of prognosis between different unsupervised clusters of patients, implicating it as a possible prognostic marker, although it should be mentioned that there was a statistically significant difference in age between these clusters [35]. Moreover, IDH status was a risk factor identified as one of the prognostic classifiers with a statistically significant high hazard ratio [40]. Additionally, several studies emphasized the importance of matching metabolic pathway markers with associated genetic alterations in a stepwise fashion to predict the prognosis of GBM patients with greater accuracy [37,39]. The emphasis of ML identification on IDH1 overexpression in the progression of GBM gives credence to an antimetabolic approach, as decreasing the activity of this pathway could impair the growth and development of GBM tumors [42]. However, IDH1 and its role in GBM metabolism has been heavily researched in the literature and is not a new discovery based on these ML papers currently being discussed. Rather, many of these papers used IDH1-positive samples as a starting point for further analysis with ML algorithms.

Various alternate metabolic markers of importance are the levels of dysregulated amino acids. These amino acids have been identified as a product of activated or deactivated metabolic pathways in GBM to increase nutrient availability for tumors [34]. Many of those amino acids were discovered in previous experiments and then analyzed by ML in these studies to differentiate patients’ glioma grading, thereby ascertaining a method for a more precise diagnosis [34,42]. In fact, Firdous et al. found that their diagnostic study utilizing an extra tree classifier, logistic regression integration, and random forest algorithms had greater predictive accuracy than any other previous studies of ML algorithms on the identification of metabolic markers in tissue or liquid-based biopsies [34].

A study conducted by Zeng et al. found that UDP glucose phosphorylase-2 (UGP2) was an upregulated enzyme that exhibited a significant effect on the prognosis of GBM. They identified this marker using a random survival forest algorithm, which is a type of supervised ML method. The overexpression of UGP2 was correlated with a worse prognosis and a higher grade of pathology. As a result, UGP2 may be a useful prognostic marker for GBM patients [38].

In addition, Kałuzińska et al., utilized multiple SVM algorithms to classify the top genes present in multiple types of cancers, including GBM. The team concluded that WWOX-dependent biomarkers PLEK2 and GCSH are possible GBM biomarkers and should serve as a triad along with RRM2. Further investigation is needed pertaining to PLEK2 and GCSH to analyze their prognostic accuracy and ability to differentiate between GBM versus alternative gliomas [36].

Lastly, several independent studies have identified EMP3 as a prognostic gene for high-grade gliomas [35,40]. It has been shown to function as a reliable indicator for prognosis at the mRNA level [40]. In fact, EMP3 was the only gene identified in more than one study. As such, further research involving this genetic marker has the potential to improve the prognostic process for patients diagnosed with glioblastoma.

## 5. Future Directions

Machine learning has the ability to greatly improve the prognostic and diagnostic capabilities of GBM. However, an integration of ML algorithms for biomarker detection combined with radiomics-based tumor imaging will be necessary to ascertain the greatest level of accuracy and precision [21]. By analyzing the characteristics of the tumor such as shape, size, and texture, radiomics can provide valuable information on the tumor’s current state and progression. Combining the two ML algorithms to analyze the quantitative data from both imaging and biomarkers could improve disease outcomes, once perfected, at a rate higher than any one method alone.

Overall, our findings highlight the importance of further research in this evolving field in order to fully grasp the potential of ML in the diagnosis and prognosis of GBM. Advancements in this area may significantly enhance patient care and treatment outcomes for individuals affected by this devastating disease in the future.

## 6. Conclusions

Machine learning is a cutting-edge technology that analyzes data and makes predictions or decisions using algorithms and statistical models. It is a formidable research tool and has the potential to completely change how complex diseases such as glioblastoma are studied and understood. The goal of machine learning is to recognize and categorize unknown data samples using training data. The studies we reviewed have found novel insights into the mechanisms of GBM and identified potential biomarkers for diagnosis and prognostication by utilizing this technology in the study of GBM metabolism.

Arguably one of ML’s most significant advantages is its ability to adapt and improve over time as it processes more data, making it ideal for dealing with complex and dynamic tasks. This is particularly useful for assignments that traditional rule-based systems are unable to manage. Additionally, machine learning can automate tasks that would normally require human intervention, increasing efficiency, and decreasing error rates. This can lead to cost savings, increased productivity, and more accurate decision making.

Conversely, machine learning has drawbacks that must be considered despite its benefits. One significant shortcoming is the requirement for large quantities of high-quality training data, which can be expensive and challenging to come by. Furthermore, it can be challenging to understand how ML models make decisions and how to optimize them because the results can be ambiguous and difficult to interpret. It is also important to remember that the effectiveness of ML models depends considerably on the caliber of the data, as well as the specific task it is assigned. Therefore, it is essential to take these factors into account when implementing ML into medical research.

GBM is a complicated disease with a limited understanding of the underlying biological mechanisms, making diagnosis and treatment challenging. The use of ML algorithms has demonstrated incredible promise in the enhancement of diagnostic and prognostic capabilities for GBM patients; however, a consensus on even a handful of biomarkers discovered with ML algorithms has not yet been made. Many researchers are still exploring this new field and there is still much to be learned. Despite the challenges and limitations, the potential of ML in the study of GBM metabolism is clear.

## Figures and Tables

**Figure 1 metabolites-13-00161-f001:**
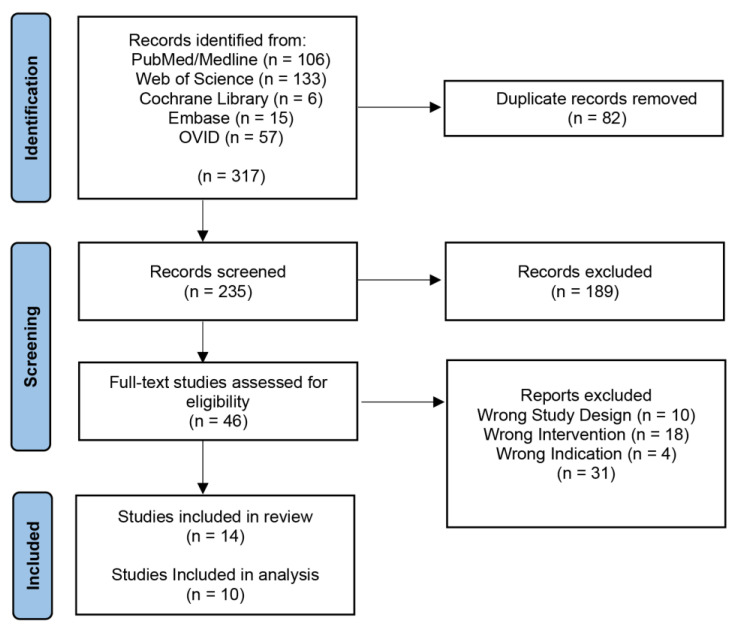
PRISMA Diagram. PRISMA (Preferred Reporting Items for Systematic Reviews and Meta-Analyses) diagram representing the screening process.

**Figure 2 metabolites-13-00161-f002:**
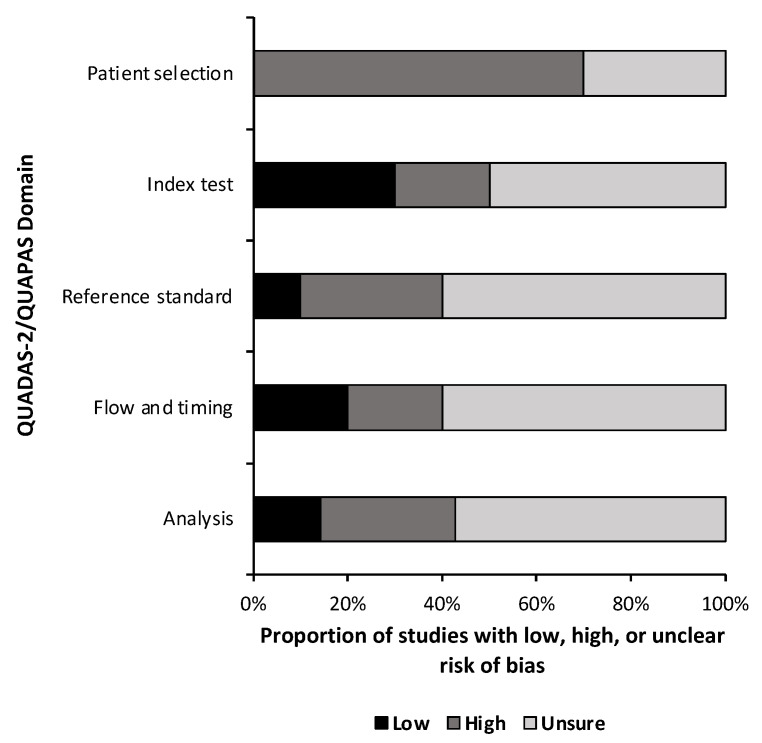
Risk of Bias. Risk of bias assessment summary based on quality assessment of diagnostic accuracy studies 2 (QUADAS-2) for diagnostic studies and the quality assessment of prognostic accuracy studies (QUAPAS) for prognostic studies.

**Figure 3 metabolites-13-00161-f003:**
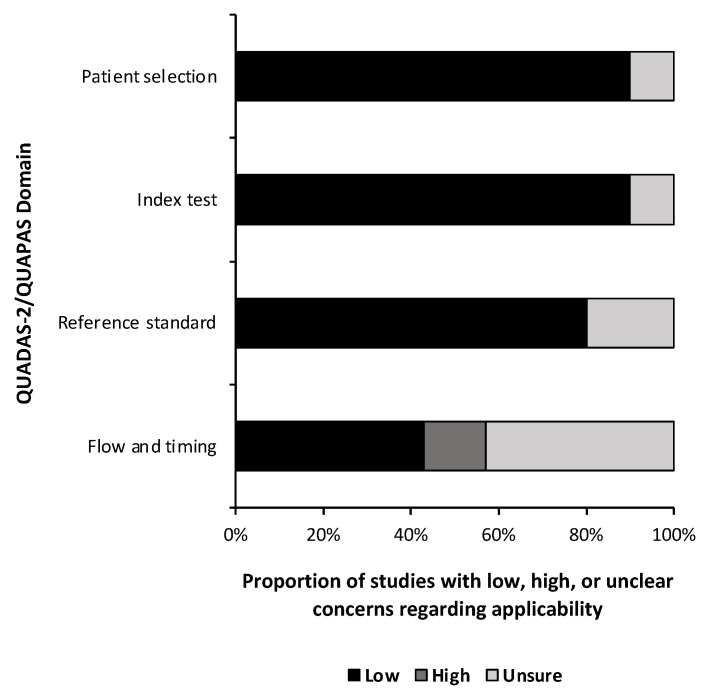
Risk of bias. Risk of bias applicability assessment summary based on quality assessment of diagnostic accuracy studies 2 (QUADAS-2) for diagnostic studies and the quality assessment of prognostic accuracy studies (QUAPAS) for prognostic studies.

**Figure 4 metabolites-13-00161-f004:**
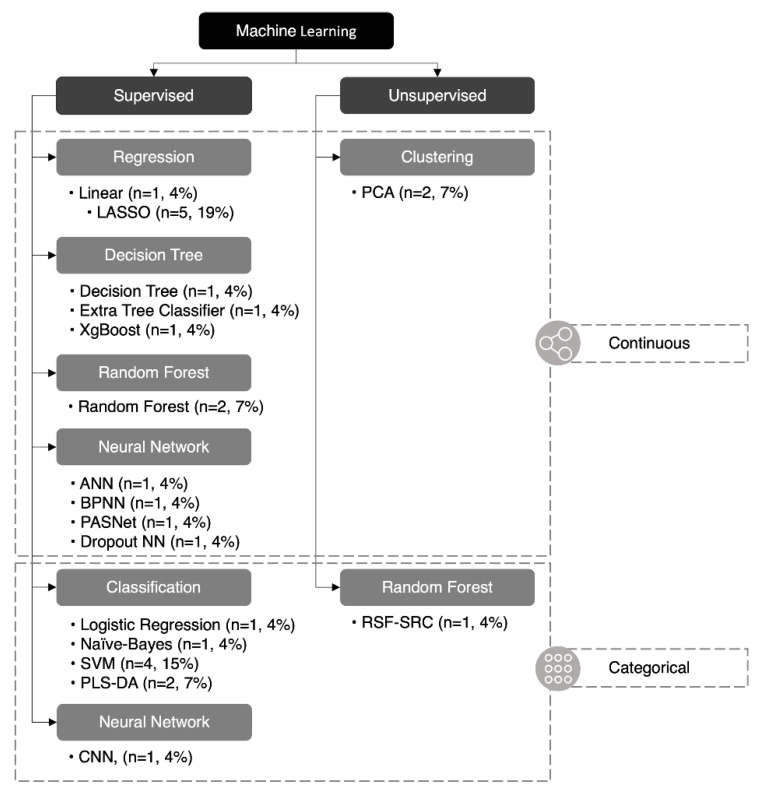
Classification of machine learning algorithms. Twenty-seven machine learning algorithms reported in this paper were classified: 24 were supervised (89%) and three were unsupervised (11%), 19 were continuous (70%), and eight were categorical (30%). ANN (artificial neural network), BPNN (backpropagation neural network), CNN (convolutional neural network), LASSO (least absolute selection and shrinkage operator), NN (neural network), PASNet (pathway-associated sparse deep neural network), PCA (principal component analysis), PLS-DA (partial least-squares discriminant analysis), RSF-SRC* (random survival forest–survival regression and classification), SVM (support vector machine), XGBoost (eXtreme gradient boosting). *RSF-SRC can be unsupervised or supervised depending on the variation.

**Table 1 metabolites-13-00161-t001:** Data extracted from ten studies were included for analysis. ACE (atlas correlation explorer), ANN (artificial neural network), BPNN (backpropagation neural network), AUC of ROC (area under the curve of the receiver operating characteristic curve), CGGA (Chinese glioma genome atlas), CNN (convolutional neural network), GEO (gene expression omnibus), LASSO (least absolute selection and shrinkage operator), NN (neural network), PASNet (pathway-associated sparse deep neural network), PCA (principal component analysis), PLS-DA (partial least-squares discriminant analysis), RSF-SRC (random survival forest–survival regression and classification), SVM (support vector machine), TCGA (The Cancer Genome Atlas), XGBoost (eXtreme gradient boosting).

Study	Country	Type	Experimental n/Control n (Total n)	Database(Location)	Category	Classification	Type of ML	Accuracy	AUC of ROC	Identified Metabolic Makers	Sample Origin
Ishwar [32] (2022)	Canada	Diagnostic	14/23 (37)	Original	Supervised	Categorical	PLS-DA	98.38%	0.957	Immune checkpoint markers: PDL1 and CTLA-4 in GBM Natural killer cell circulating immune vesicles	Serum
Unsupervised	Continuous	PCA		
Supervised	Continuous	ANN	100%	1.000
McInerney [33] (2022)	Switzerland	Both	n/a	TCGA	Supervised	Continuous	ACE (Linear Regression)			Prognostic: TSPYL2, JAKMIP1, CIT, TMTC1 Diagnostic: MINK1, PLEKHM3, BZW1, RCF2	Tissue
Firdous [34] (2021)	Pakistan	Diagnostic	26/16 (42)	Original	Supervised	Continuous	Extra Tree Classifier	100%	0.760	alanine, glutamine, valine, methionine, N-acetyl aspartate (NAA), γ-aminobutyric acid (GABA), serine, α-glucose, lactate, and arginine	Plasma
Supervised	Continuous	Random Forest	100%	0.780
Supervised	Categorical	Logistic Regression	98%	0.860
Jia [35](2021)	China	Prognostic	154	TCGA	Supervised	Continuous	BPNN		0.865	GPX8, CCDC109B, IGFBP2, LINC00152, LOC541471, METTL7B, S100A4, EMP3, CLIC1, TAGLN2	Tissue
Supervised	Categorical	SVM		0.862
Supervised	Continuous	CNN		
Supervised	Continuous	XGBoost		0.718
Supervised	Continuous	Random Forest		0.724
Supervised	Continuous	LASSO		0.874
Kaluzinska [36] (2021)	Poland	Prognostic	n/a	Original	Supervised	Categorical	SVM		0.935	PLEK2, RRM2, GCSH, BMP4, CCL11, CUX2, DUSP7, FAM92B, GRIN2B, HOXA1, HOXA10, KIF20A, NF2, SPOCK1, TTR, UHRF1	Tissue
He [37](2020)	China	Prognostic	381	TCGA, CGGA, GEO	Unsupervised	Continuous	PCA			ACADS, ADRA2A, ALAS1, APOD, ARSF, ESRRB, FOXO3, HSPH1, KLF15, NR1H4, PCSK1, PIK3R1, RNASEL, RUFY1, SFN, SH3GLB1, SPTSSA	Tissue
Supervised	Continuous	LASSO-Penalized Cox regression		0.752
Zeng [38] (2019)	China	Prognostic	252	TCGA, GEO	Unsupervised	Categorical	RSF-SRC			UGP2, TUBB2A, FABP3, SLC17A7, NAGPA, PRKCB, DNM1, NEFM, TIMP1, ITGB1, MRC2, TAF9B, MAT2A, HSPD1, PDLA4	Tissue
Hao [39](2018)	USA	Prognostic	522	TCGA	Supervised	Continuous	PASNet		0.662	CDC42, PRKCQ, RAC1, AKT1, AKT2, AKT3, C3, CREB1, GRB2, HRAS, KRAS, NRAS, PRKACA, PRKACB, PRKACG, RAF1, and YWHAB,	Tissue
Supervised	Continuous	Logistic LASSO		0.590
Supervised	Continuous	Random LASSO		0.621
Supervised	Categorical	SVM		0.634
Supervised	Continuous	Dropout NN		0.641
Shu [40](2018)	China	Prognostic	193 original, 875 databases (1068)	Original, CGGA, TCGA, GEO	Supervised	Continuous	LASSO		0.778	Genes: WEE1, EMP3, IGFBP3Biomarker: WEE1	Tissue
Gollapalli [41] (2012)	India	Diagnostic	40/40 (80)	Original	Supervised	Continuous	PLS-DA	92.85%		haptoglobin, plasminogen precursor, apolipoprotein A-1, and M, transthyretin, cholesterol, triacylglycerol, and low-density lipoproteins	Serum
Supervised	Categorical	SVM	92.85%	
Supervised	Continuous	Decision Tree	92.85%	
Supervised	Categorical	Naïve Bayes	85.70%

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
