# Peer review of "Assessing Metabolic Markers in Glioblastoma Using Machine Learning: A Systematic Review"

_metabolites, 2023, doi:10.3390/metabo13020161_

Round 1

Reviewer 1 Report

I am really grateful for reviewing this manuscript. In my opinion, this manuscript can be published once some revisions are done successfully. This study performed a systematic review for ten articles of machine learning on glioblastoma metabolism diagnosis and prognostication. Based on the results of this systematic review, the mean accuracy and area under the receiver-operating-characteristic curve of the reviewed studies were 96% and 78%, respectively. According to the findings of this systematic review, indeed, 106 metabolic markers were identified as major predictors but EMP3 was the only predictor reported in multiple studies. I would argue that this systematic review is a rare achievement. However, I would like to make two suggestions to improve this manuscript. Firstly, I would like to suggest the authors to explain about machine learning models in the section of Introduction, which would help readers to understand the article much better. Secondly, I would like to suggest the authors to elaborate on the issue of “an integration of machine learning algorithms for biomarker detection combined with tumor imaging” in Conclusion, which is a major issue among experts now. 

Reviewer 2 Report

Conclusion section should be more elaborate.

Directions for the future study should be included as a separate section.

Section 3.4,Authors mentioned 24ML algorithms were found. Provide overview for each of the ML algorithms

Figure 4 is bit confusing, why Random forest and Neural networks are falling under Unsupervised learning?

What are the pros and cons of using machine learning in Glioblastoma Metabolism Diagnosis and Prognostication

Authors should expand the study by collecting more references.
